**Field-scale water balance closure in seasonally frozen conditions**
X. Pan[1,2], W. Helgason[3,1], A. Ireson[4,1], H. Wheater[1]
[1]Global Institute for Water Security, University of Saskatchewan, Saskatoon, SK, Canada
[2]State Key Laboratory of Soil and Sustainable Agriculture, Institute of Soil Science, Chinese
Academy of Sciences, Nanjing, China
[3]Civil and Geological Engineering, University of Saskatchewan, Saskatoon, SK, Canada
[4]School of Environment and Sustainability, University of Saskatchewan, Saskatoon, SK, Canada
**Abstract**
Hydrological water balance closure is a simple concept, yet in practice it is uncommon to measure
every significant term independently in the field. Here we demonstrate the degree to which the
field-scale water balance can be closed using only routine field observations in a seasonally-frozen
prairie pasture field site in Saskatchewan, Canada. Arrays of snow and soil moisture measurements
were combined with a precipitation gauge and flux tower evapotranspiration estimates. We
consider three hydrologically distinct periods: the snow accumulation period over the winter, the
snowmelt period in spring, and the summer growing season. In each period, we attempt to quantify
the residual between net precipitation (precipitation minus evaporation) and the change in field
scale storage (snow and soil moisture), while accounting for measurement uncertainties. When the
residual is negligible, a simple 1D water balance with no net drainage is adequate. When the
residual is non-negligible, we must find additional processes to explain the result. We identify the
hydrological fluxes which confound the 1D water balance assumptions during different periods of
the year, notably blowing snow and frozen soil moisture redistribution during the snow

accumulation period, and snowmelt runoff and soil drainage during the melt period. Challenges associated with quantifying these processes, as well as uncertainties in the measurable quantities, caution against the common use of water balance residuals to estimate fluxes and constrain models in such a complex environment.

**Keywords: water balance closure; field scale; seasonally frozen soils; hydrological processes**

## 1 INTRODUCTION

Water balance closure has been described as the holy grail of scientific hydrology (Beven, 2006). Beven suggests that the most important problem in hydrology in the 21st Century is providing the techniques to measure integrated fluxes and storages at useful scales. In the current paper, we define the problem of water balance closure as that of independently quantifying each term in the water balance equation, such that the changes in storage within a specified domain and over some time interval are adequately balanced by the net fluxes into/out of that domain over the same time interval. As simple as this concept is, it has proven to be extremely hard to achieve in field studies. For example, Mazur et al. (2011) reported a water balance closure study for a well-characterized, intensively monitored artificial catchment, and were unable to close the water balance due to their inability to quantify evapotranspiration and changes in storage. Natural heterogeneity of both water fluxes and moisture states, which can vary at spatial and temporal scales that are beyond (or beneath) our measurement capabilities, can make the task of observing complete water balance closure seem like an enigmatic pursuit.

In this paper we present a case study from a heterogeneous pasture site in the Canadian prairies, where we have quantified the various components of the water balance at the field scale, and critically examine some of the simplifying assumptions which are often invoked when applying water budget approaches in applied hydrology. The Canadian prairie region lies in the southern part of the provinces of Alberta, Saskatchewan and Manitoba, and makes up the northern portion of the Great Plains region of North America. The hydrology of this region is markedly influenced by the regional climate and geology, and at first glance appears to have a relatively simple water balance. Much of the rainfall occurs during the growing season and is consumed by evapotranspiration, resulting in very little surface runoff. Extensive past glaciations have blanketed

the region with a thick compacted till which has very low permeability (Keller et al., 1989), resulting in relatively small interactions between the surface water and the underlying groundwater regime (van der Kamp and Hayashi, 2009). As such, the water balance in this region is conceptualized to be dominated by vertical exchanges of precipitation and evapotranspiration between the soil and the atmosphere.

However, certain characteristics of the prairies region also make the hydrology complex and are likely to confound simple 1-D assumptions regarding the water budget. The region is seasonally frozen, with long winters (4-6 months), featuring many cryosphere-dominated hydrological processes. Approximately one third of annual precipitation is snowfall, which is subject to extensive wind redistribution throughout the landscape (Pomeroy et al., 1993; Pomeroy and Li, 2000) resulting in a spatially variable water input. During the spring melt, spatially variable surface albedos and heat advection from snow-free to snow-covered areas can cause differential rates of snow melt (Shook et al., 1993; Liston, 1995). Moreover, infiltration into frozen soil has complex dependencies upon the antecedent moisture, the rate of melt, and local topography (Gray et al., 2001), resulting in a highly variable spatial infiltration pattern (Hayashi et al. 2003, Lundberg et al., 2016). Due to these factors, the annual snowmelt event typically produces 80% or more of the annual local surface runoff (Gray and Landine, 1988). The hydrological complexity of the landscape is also largely influenced by glacial and post-glacial geomorphological processes which have imparted a tremendous degree of heterogeneity. Morainal deposits, comprised of a variable mix of soil textures, are often topographically indeterminate and consist of areas which are internally drained and infrequently contribute to stream flow (Zebarth and de Jong, 1989; Shaw et al., 2012).

While observations of all of the hydrological fluxes and states at large, i.e. useful (Beven, 2006), scales are desirable, current measurement approaches do not yet fully permit this. The evaporative flux can be measured over reasonably large scales (on the order of hundreds of meters) using the eddy covariance technique, whereas the soil moisture status and bottom drainage fluxes can generally only be measured at point scales. Recent advances in remotely sensed soil moisture, such as the ground-based cosmic ray neutron probe (Zreda et al., 2008) or satellite-based sensors such as those used by the Soil Moisture and Ocean Salinity (SMOS) mission (Kerr et al., 2010) or the Soil Moisture Active Passive (SMAP) mission (Entekhabi et al., 2010), can retrieve soil moisture estimates over hundreds of meters to tens of kilometers. However, these observations are limited to the near-surface, and need to be depth-scaled to the root-zone to be suitable for water balance studies (Peterson et al., 2016). Adequately capturing field-scale variability using point scale measurement techniques requires a large number of samples (e.g. Grayson and Western, 1998; Famiglietti et al., 2008; Brocca et al., 2010).

The objective of this paper is to explore how well the water balance can be closed using only routine field observations in a seasonally frozen environment. We use a well-instrumented field site to quantify the magnitude of the water balance components as they vary across three distinct seasons in the prairies at field scale. We start with a conceptual model of all of the dominant hydrological processes active at the site, from which we construct a water balance equation. We designed a simple field experiment to measure the components of the water balance that can be measured in a routine, if labor intensive, manner at field scale. We performed an uncertainty analysis on each measurement, accounting for instrument error and sampling error. If the non-measured terms are negligible, i.e. within sampling and instrumentation errors, we are left with a 1D water balance where net precipitation (precipitation minus evapotranspiration) is adequately

well balanced by changes in storage only. If the non-measured terms are non-negligible, there will
be a non-zero water balance residual. We interpret these residuals for the different seasons. We
evaluate the validity of treating the problem as one-dimensional in different seasons, and the value
of using water balance residuals to estimate fluxes and constrain models.
**2 METHODS**
**2.1 Field-scale water budget**
We consider field scale to represent an area of the order of 500 m x 500 m, from the ground surface
to a depth of 1.6 m. This was the depth range that we were able to install neutron probes to monitor,
and is deep enough to capture all of the significant soil moisture dynamics at our site. At this scale,
storage terms include surface storage, $\Delta S_s$, which includes snow and ponded water, and subsurface
(vadose zone) storage, $\Delta S_v$, which is liquid and solid (ice) soil moisture integrated over the root
zone (taken to be 1.6 m). The field-scale vertical water balance can hence be expressed for the
surface as

$$\Delta S_s = P - E_S - I - G - O \tag{1}$$

for the subsurface as

$$\Delta S_v = I - E_B - D \tag{2}$$

and for the overall field scale as

$$\Delta S_T = \Delta S_S + \Delta S_v = P - E - O - G - D \tag{3}$$

where all terms are in units of mm, and $P$ is precipitation (solid and liquid phases), $I$ is infiltration,
$E$, $E_S$, and $E_B$ are total evaporation ($E = E_S + E_B$), surface evaporation (including free water

evaporation and snow sublimation) and subsurface evaporation (including soil evaporation and plant transpiration), respectively, $O$ is surface runoff leaving the field domain, $G$ is net drifting snow over the field domain, and $D$ is vertical soil drainage at 1.6 m depth.

The water table is located 3-5 m below ground surface depending on location (the water table is shallower in topographic depressions) and time of year (the water table is shallowest in the earlier summer after the melt period). Water table dynamics are modest, but there will be lateral saturated flow processes occurring. Since the saturated zone is well below the domain of our water balance, here we only consider vertical drainage from the base of our soil layer. Lateral unsaturated subsurface flow may occur at local scales due to changes in soil properties, but we do not expect these fluxes to be significant at field scale. Hence, lateral subsurface fluxes are neglected.

In seasonally-frozen environments where winters are long and cold, processes in the summer and winter are markedly different. A water year in this region is typically defined as from November to October, such that snow accumulation and melt occur within the same water year. Annual water balances are useful, but do not elucidate the important seasonal processes – in particular the storage dynamics. For a more rigorous analysis, here we examine the water balance over three distinct seasons: *The snow accumulation period* starts from the first killing frost ($<-2°C$), and ends at the beginning of snowmelt; t*he melt period*: in a typical year this starts from the peak snowpack, and ends when the ground is completely snow free, typically 2-4 weeks sometime in March, April or May; and *the growing season*: starts from the end of the melt period, and ends with the onset of the snow accumulation period, roughly May to October.

In each period the nature of the individual components of the water budget is different. For example, in the snow accumulation period, surface storage occurs as snow; in the growing season, if it exists at all, it is as ponded water in ephemeral ponds which tend to dry out in early summer;

and in the melt period, it is a transition between these two. Snow drift, runoff and evaporation are typically only significant in the snow accumulation season, the melt period, and the growing season, respectively. This will be discussed in detail in Section 4.

## 2.2 Description of study site

The instrumented field site (51° 22′ 54″ N, 106° 24′ 57″ W) lies within a gauged sub-basin (Fig. 1) of the Brightwater Creek watershed, which is a sub-basin of the South Saskatchewan River Basin. The gross area of the sub-basin defined by the Water Survey Canada gauge (05HG002) is 900 $km^2$, while the effective basin area, or that which would be expected to contribute flow to the main stream channel during a flood with a return period of 2 years (Martin 2001), is just 282 $km^2$ (Fig. 1a). Mean annual precipitation is about 330 mm (2009-2014), of which about 70 mm typically falls as snow. Mean annual yield via streamflow in the Brightwater Creek watershed is $4.95 \times 10^6$ $m^3$ (1983-2013), equivalent to 5.5 mm over the gross drainage area or 17.6 mm over the effective drainage area. Annual runoff is therefore small by either measure. Streamflow is intermittent, and in most years only occurs following snowmelt. The mean temperature in January and July is -12.9°C and 18.8°C, respectively. The regional landscape consists of gently sloping glacio-lacustrine plains surrounded by moraine deposits that have a rolling knob- and kettle-type topography (Miller et al., 1985). The soils in the region are mainly Solonetzic and Chernozemic, and are mapped as Bradwell and Asquith Associations as described by Ellis et al. (1968).

This local study area (500 m × 500 m) is located within a ~700 ha grazing pasture, which is surrounded by fields cultivated in annual crops. Within the instrumented region, the topography is undulating with a range in elevation of approximately 5 m. Vegetation consists of various

wheatgrasses (*Agropyron* sp.) and needle grasses (*Stipa* sp.) with patches of western snowberry (*Symphoricarpos occidentalis*), commonly referred to as buckbrush. Brush and grass communities are interspersed in a spatial pattern on the order of 10's of meters. The texture of the soils within the study area ranges from loam to clay loam.

**2.3 Instrumentation**

A variety of measurements were used to characterize the field scale water balance from November 1, 2012, reported here until October 31, 2014. The total evaporation flux, *E,* (mm) was obtained using the eddy covariance technique. This consisted of a Campbell Scientific CSAT3 sonic anemometer and a Campbell Scientific KH20 krypton hygrometer mounted on a scaffold tower located in the center of the study area. The instruments were mounted at a height of 4.85 m above ground, and had a representative measurement fetch of approximately 500 m (Burba, 2013). Raw data were collected at a rate of 10 Hz, and latent heat fluxes ($Q_E = \lambda E$, where $\lambda$ is the latent heat of vaporization or sublimation) and sensible heat flux ($Q_H$) fluxes were calculated using Licor EddyPro software (www.licor.com/eddypro). Gaps in the flux data were filled using the Kalman filter and a dynamic linear regression for recursive parameter estimation developed by Young and coworkers (Young, 1999; Young and Pedregal, 1999; Young et al., 2004), based on the relationship between latent heat flux, available energy, and vapour pressure deficit. This gap-filling approach was evaluated and recommended by Alavi et al. (2006) for filling gaps in latent heat flux data.

The available energy, consisting of the net radiation flux ($Q_{NR}$) and the ground heat flux ($Q_G$) was measured at two locations within the eddy-covariance measurement footprint: representing

grass and brush surfaces. At the scaffold tower (grass surface), net radiation fluxes were measured

using a Kipp & Zonen CNR1 4-component radiometer; whereas, net radiation fluxes at an auxiliary

tripod (brush cover) located approximately 100 m from the scaffold were measured with a

Hukseflux NR01 4-component radiometer. At both locations 2 heat flux plates (Radiation Energy

Balance Systems model HFT3) were installed at a depth of 8 cm, and were laterally separated by

~1 m. In order to calculate energy storage in the soil layer above the heat flux plates $(\Delta S_G)$, a single

volumetric water content sensor (Campbell Scientific CS650 dielectric permittivity sensor) was

installed at 5 cm depth, and a pair of averaging thermocouples were installed at 4 cm depth. The

available energy, calculated as the net radiation flux minus (plus) the amount of energy transferred

into (from) the soil, was averaged between the 2 locations.

The energy balance closure ratio (*EBR*),

$$EBR = \frac{\sum(Q_E + Q_H)}{\sum(Q_{RN} - Q_G - \Delta S_G)}, \tag{4}$$

was evaluated for each day of the growing season, which gave an average closure fraction of 0.72

and 0.74 in 2013 and 2014, respectively. These biases were corrected by forcing energy balance

closure using the measured Bowen ratio (*cf.* Twine et al., 2000; Barr et al., 2012) on a daily basis,

which increased the measured seasonal evaporation fluxes by 39% and 35% in 2013 and 2014,

respectively. Biases for the other seasons were not corrected since the evaporation fluxes over the

frozen ground surface were very small, and the turbulent heat fluxes are much more uncertain over

snow (Helgason and Pomeroy, 2012).

Precipitation (mm) was measured by a Geonor T200-B weighing gauge. Biases in solid

precipitation (i.e. snow) measurements were corrected for undercatch using a wind speed-catch

efficiency relationship (Smith, 2008), and for liquid precipitation (i.e. rain) we assume a catch
efficiency of 95% for all rainfall measurements (Devine and Mekis, 2008). Precipitation bias-
correction leads an increase in measured precipitation of 19% and 13% in 2013 and 2014,
respectively.
Root zone soil water content and snowpack depth and density were measured at point locations
in a crosshair pattern, comprising two perpendicular transects, centered on the flux tower (shown
in the upper right corner of Fig. 1a). Water content was measured by a down-hole neutron moisture
meter, model CPN 503DR Hydroprobe (CPN International Inc., Concord, CA). The blue pins are
the neutron probe reading locations installed in June, 2012, and the yellow pins show new locations
added in summer 2013. Volumetric soil moisture content (liquid water + ice) was measured at
depth intervals of 0.2 m, from 0.2 m to 1.6 m below ground. Due to the problem of surface loss of
neutrons, no readings shallower than 0.2 m were taken, meaning we may underestimate the
changes in water content at the top of the soil profile. The change in soil water storage, $\Delta S_v$, was
calculated as the difference between any two moisture surveys, which were conducted with a time
interval of around 2 weeks in the unfrozen period, and 2-3 times during the frozen period.
Snowpack distribution along the long transect was investigated with a series of snow surveys
during the snow covered period in late winter/spring of 2013 and 2014. Snow depth was measured
at a distance interval of about every 2.0~3.0 m, and snow samples were taken from the neutron
probe locations, i.e. every 50 m, using a core sampler (ESC30, Environment Canada, Canada) to
determine snow density and calculate snow water equivalent (SWE). The change in snow water
storage, $\Delta S_s$, was calculated as the difference in the mean SWE between any two sampling dates.
The topography of the long transect from northwest to southeast is shown in Fig. 1b.
Water table depths were monitored using piezometers, screened (33-cm in length) at a depth of
around 5.5 m below ground, with level loggers (Solinst, Model 3001) at three locations along the
northwest-southeast transect (#1, #2 and #3 in Fig. 1b). The one closest to the flux tower started
collecting data on July 17, 2012, and the other two started on October 7, 2013. The measured water
table depth was corrected for changes in barometric pressure, measured at the flux tower, using
the graphical method for estimation of barometric efficiency proposed by Gonthier (2007).
Soil temperature was measured using Stevens Hydro-probes at three profiles, co-located with
the piezometers. At profile #1 (Fig. 1b) five probes were installed, at depths of 0.05 m, 0.2 m, 0.5
m, 1.0 m and 1.5 m below ground, and at the other two profiles seven probes were installed, at
depths of 0.05 m, 0.2 m, 0.5 m, 0.75 m, 1.0 m, 1.3 m and 1.6 m below ground. Measurements were
recorded every 30 minutes. The depth of the freezing front as a function of time was calculated by
interpolating the 0°C line from the soil temperature measurements.
**2.4 Uncertainty assessment**
We performed a quantitative uncertainty analysis of all of the measured terms in our water balance
assessment. We expect uncertainties in our measurements of precipitation and evapotranspiration
to be dominated by measurement errors. Conversely, we expect uncertainties in our measurements
of soil moisture and snowpack to be dominated by sampling errors. These four terms make up a
naïve, 1D, water balance, where the net precipitation (defined as $P - E$) equals the total change
in storage ($\Delta S_s + \Delta S_v$). The water balance residual, $R$, is given by:

$$R = P - E - \Delta S_s - \Delta S_v. \tag{5}$$

If $R$ is negligible, we can say the 1D water balance is appropriate. If $R$ is significantly larger than
zero in any period, we must expect one or more other fluxes from Eqn 1-3 to be significant in that
period. We seek to quantify an error bound for each term, $\pm\varepsilon$, and combining these errors by
summing in quadrature to establish an error bound for the residual, $\varepsilon_R$, given by (Coleman and
Steele, 1989):

$$\varepsilon_R = \sqrt{(\varepsilon_P)^2 + (\varepsilon_E)^2 + \left(\varepsilon_{S_s}\right)^2 + \left(\varepsilon_{S_v}\right)^2}. \tag{6}$$

The error bounds for each measurement term are described in the following paragraphs. In all
cases, we concentrate on quantifying the largest, most dominant, source of uncertainty.
*Precipitation.* With respect to the weighing gauge used here (Geonor T-200b), instrument error
is actually quite small, i.e. the manufacturer provides an accuracy estimate of 0.1% full scale
(which is only 0.6 mm). Similarly, Duchon (2008) presents an example validation of the factory
calibration equation demonstrating that minor calibration errors typically introduce small biases
(<1%) which occur at small or large bucket volumes. However, comparatively large biases can
occur due to wind-induced undercatch of solid precipitation (Goodison et al., 1998). In this study,
we correct for this bias using the relationship provided by Smith (2008), which predicts the gauge
catch efficiency as a function of windspeed. Details on the correction method, and its relative
importance in the prairie environment can be found in Pan et al. (2016). Owing to experimental
difficulties in developing a catch efficiency equation, there is considerable (but not quantified)
uncertainty which must be introduced when applying the correction factor (this is inferred by the
relatively large scatter in Fig 4 and 6 of Smith (2008)). Thus we consider the dominant uncertainty
of the winter precipitation measurements to be the wind induced undercatch (bias) $\pm$ a random

uncertainty associated with the applied correction factor. In order to estimate the latter, we obtained

the original data from Smith (2008) and calculated the prediction intervals (95%) on the catch

efficiency equation. These were then used to estimate the random error introduced by applying the

correction factor to each snow event for the current study. The cumulative random error for each

snow accumulation period was calculated by summing all of the individual event errors in

quadrature. Undercatch errors are much larger for solid precipitation as compared with liquid

precipitation, thus all rainfall values are corrected for an undercatch 5% bias (Pan et al., 2016). On

an annual basis, these systematic corrections results in an increase in precipitation of 47 mm in

2013 and 44 mm in 2014. It is hard to rigorously quantify any additional random errors in the

precipitation measurement, so we have assumed random errors of 10% of daily precipitation.

*Evaporation*. Measurements of evaporation using the eddy covariance technique may be subject

to random measurement error as well as systematic errors due to instrument limitations, unmet

theoretical assumptions, or processing issues (Richardson et al., 2012). In this study we deal with

random errors in an overly simplistic manner by assuming that they are 10% of the daily E. While

this may be the approximate order of magnitude for water vapour flux random errors (e.g. Moncrief

et al., 1996; Litt et al., 2015), the approach is only justified in this case by the fact that the random

errors are dwarfed by the systematic errors. While some systematic errors are corrected for during

flux processing (e.g. sensor separation, density fluctuations, high frequency losses, etc.) it is clear

that there are other systematic errors that are not accounted for. We deal with these by forcing

energy balance closure (Eqn. 4), which on an annual basis, resulted in an additional 80 mm of

evapotranspiration in 2013 and 95 mm in 2014.

*Snow*. The accuracy of estimating the areal snow water equivalent depends on the measurement

accuracy, and the sampling uncertainty. The snow survey equipment used in this study, i.e. the

ESC30 snow coring tube (Farnes et al., 1980; dimensions also given in Kinar and Pomeroy, 2015),
has a relatively large 30 cm$^2$ cutter area, which reportedly allows it to measure the snow density
within 1% of the true value (Farnes et al., 1983; Goodison et al., 1987). Ultimately this depends
on ability to cut the snow sample and retain it in the tube, transfer cleanly to a sample bag, and
accurately measure the mass. Generally it is accepted that these errors can be minimized by an
experienced surveyor. However, a far more significant challenge is to accurately assess the mean
SWE by collecting a finite number of samples from heterogeneous snow field. The confidence
associated with this estimate is rarely reported. In this study, we use a bootstrap sampling technique
(described below) to assess the standard error of the mean snow depth and mean snow density,
which were propagated to obtain the standard error (*SE*) for the mean areal SWE. The 95%
confidence intervals were then calculated as 1.96 x *SE*. An important assumption of this approach
is that the snow density samples, collected on 50 m spacing, can be considered random. For
shallow prairie snowpacks, random behaviour is found after length scales of around 30 m (Shook
and Gray, 1996).
*Soil moisture.* Soil moisture measurements obtained using the neutron thermalization technique
are subject to instrument errors, calibration errors, depth integration errors, and spatial sampling
errors (Vandervaere et al., 1994). Similar to the estimation of SWE, we consider limited sampling
in space to be the largest form of uncertainty in estimating soil moisture changes. Some of the
instrument and calibration errors are minimized when changes in soil moisture are of interest rather
than the total soil moisture storage (Vandervaere et al., 1994). In order to calculate the 95%
confidence intervals around the spatial mean soil moisture change we used a bootstrap resampling
technique (e.g. Cosh et al., 2004) in which the soil moisture change was resampled 5000 times
(with replacement). The standard error of the areal mean was obtained as the standard deviation of

all of the bootstrapped mean estimates. In cases where the data were normally distributed, the 95% confidence interval was taken as ±1.96 x *SE*. In the event where the data were not normally distributed, the confidence intervals were found using a percentile method.

## 3 RESULTS AND DISCUSSION

The hydrological conditions over the water years 2013 (November 1, 2012 to October 31, 2013) and 2014 (November 1, 2013 to October 31, 2014) are shown in Fig. 2, and the quantified water balance components for each season are presented in Table 1. Total annual precipitation was 302 mm (2013) and 386 mm (2014). In both years, snow accumulation started around the beginning of November and snowmelt was complete by the end of April. The undercatch-adjusted snowfall was coincidentally the same in both years: 72 mm. Rainfall was 230 mm (2013) and 314 mm (2014). The sum of evaporation, transpiration, and sublimation was 285 mm (2013) and 368 mm (2014). Both years had low measured sublimation: 14 mm (2013) and 10 mm (2014). Most evapotranspiration occurred in June, July and August. Water year 2013 was typical for the region in that soil moisture was recharged following snowmelt, and then experienced drying over the summer months as *E* exceeded precipitation. Water year 2014 experienced a wet May-June, so that soil moisture decreases were delayed to the latter part of the summer. In the following sections, the dominant hydrological processes and water balance closure for each of the three seasons are described.

## 3.1 Snow accumulation period

Lateral exchange of blowing snow during the snow accumulation period is an important
characteristic of open prairie environments, and it is essential to account for this in any water
balance study. The sub-field scale distribution of snow within our instrumented field was strongly
affected by trapping of snow at the fenced tower (location M0) and within brush vegetation (e.g.
location S3). The spatiotemporal distribution of snow along the long transect (Fig. 1) is shown in
Fig. 3. Topographic effects can also play a role in snow redistribution, but here were negligible.
The phenomenon also operates at scales larger than our field site. Whether this results in a net
influx or efflux to a particular site generally depends on the relative height of the local and
surrounding vegetation, which can trap snow (Pomeroy et al., 1993). Taking sublimation loses into
account, Table 1 shows that in 2013 there was markedly more snow on the ground (72 mm SWE)
than there was $P$-$E$ (58 mm), while in 2014 the two matched closely (both 62 mm). This suggests
that in 2013 there was a net contribution of blowing snow to the pasture, meaning that the
vegetation within the pasture (grass and shrubs) were more effective at trapping blowing snow
than adjacent cropped fields (shorter stubble, usually less than 15 cm). In 2014 the net effect of
blowing snow appears to have been negligible, which is to say that the influx and efflux of blowing
snow were balanced. The continuous snow water balance through this period is shown in Fig. 4.
Here it can be seen that random errors in the accumulated precipitation are very small compared
with the systematic errors associated with undercatch of solid precipitation, and the variable
influence of blowing snow.
During the snow accumulation period the soil freezes progressively from the surface downwards.
The maximum freezing depths were 1.3 m (2013) and > 1.6 m (2014), as shown in Fig. 2d. The
reasons for the differences in freezing depth are a combination of multiple factors, which are
beyond the scope of this study to determine. The important point from a water balance perspective

is that in both years there was a non-negligible increase in soil water content over the winter (24 ±11 mm in 2013 and 10 ± 5 mm in 2014, Table 1). Figure 5 shows the change in root zone water content over the winter (from before soil freeze up, to just before the soil thawed), from all available neutron probe measurements. There were increases in water content over the winter, with larger increases nearer to the surface. The variability of change in soil water content also increases significantly nearer the surface, which implies that the wetting process is non-uniform across the field. Under frozen conditions the water content of the root zone can potentially increase due to infiltration of mid-winter snow melt events (uncommon, but not unheard of in this environment), or by upward migration caused by freezing induced hydraulic gradients (Hoekstra, 1966; Gray and Granger, 1986). It should be noted that the first set of soil moisture measurements in 2013 did not coincide with the peak SWE survey and the end of the accumulation period (April 8th), but rather occurred on April 22nd so it is possible that some melting snow had infiltrated by that date. During the defined snow accumulation periods, there were no observations of mid-winter melt events in this period (i.e. the temperature did not significantly rise above zero), so we do not believe significant infiltration occurred, and upward moisture redistribution is a more plausible explanation. Note that the water table dropped through the winter (Fig. 2), which could also be partly due to upward water migration (Gray and Granger, 1986, Butler et al., 1996, Iwata et al., 2010).

In terms of the total water balance residual, $R$, (Eqn. 5) we find large residuals (63 ± 15 mm in 2013, 13 ± 13 mm in 2014, Table 1), due to soil moisture redistribution and blowing snow, which invalidate the naïve 1D water balance in this period. In both cases, measurements of changes in storage can be made reliably, but are subject to significant uncertainty associated with sampling errors, which have likely not been given enough attention in the past. We cannot easily measure

the fluxes (namely soil drainage, $D$, which must be negative, and blowing snow, $G$) needed to close a water balance which would corroborate these changes in storage. If the surface water balance (Eqn. 1) is considered separately, the residual $R_s$ becomes smaller ($39 \pm 10$ mm in 2013, $0 \pm 7$ mm in 2014). This approach can be justified if there is no infiltration in the snow accumulation period, in which case we attribute the imbalance to the addition of snow blowing onto the field.

**3.2 Melt period**

The observed timing and magnitude of snowmelt and discharge in Brightwater Creek (measured at gauging station 05HG002) in 2013 and 2014 are compared in Fig. 6. Runoff from our field site may or may not have directly contributed to this watershed scale discharge (see the effective area in Fig. 1a), but the local infiltration/runoff behavior can still explain the differences seen at the larger scale. The timing of peak discharge in both years is consistent with the timing of the depletion of the snowpack by melting. However, the magnitude of the peak discharge in 2014 is much bigger than that in 2013. Snowpack depths were comparable (Fig. 6), but field-average SWE was significantly higher in 2013 (Table 1), which indicates there is some complex behavior in terms of the runoff generation mechanism, which we explore here. In both years, there was a large negative water balance residual, meaning melt from the snowpack exceeded the increase in soil moisture, and hence water was lost from the domain, most likely as runoff, $O$ (but possibly also as drainage, $D$). Peak SWE and melt period changes in root zone soil moisture, were measured at coincident points on the transect (Fig. 1b), and the results are shown in Fig. 7 (note that the transect was extended in 2014). In this Figure we show the peak SWE and the additional rain that fell during the melt period as inputs (positive), and the increases in soil moisture (shown as negative numbers) are shown separately for the snowmelt period ($\Delta S_{V1}$), and the subsequent snow-free soil

thaw period ($\Delta S_{V2}$), which takes considerably longer to complete (Fig. 2). The spatial patterns of snowmelt infiltration are generally consistent between years. More notable is the difference in timing of the increases in soil moisture: in 2013 all of the infiltration occurred while the soils were still frozen, whereas in 2014 there was very little infiltration into frozen soils, and most of the infiltration occurred after the soil had thawed. In 2013 there was 54 mm of infiltration of snowmelt, while in 2014 the 30 mm of infiltration was likely mainly due to rainfall during the late melt period (33 mm). In 2013 we see a runoff residual of 37±28 mm, equivalent to a snowmelt runoff ratio of 38%, while in 2014 the smaller SWE led to a larger runoff residual of 50±29 mm, equivalent to a snowmelt runoff ratio of 80%. These residuals are consistent with the observed differences in basin-scale runoff for 2013 and 2014.

To explore the marked differences in snowmelt infiltration in the two years, soil water content profiles for pre-melt, post-snowmelt, and post-thaw conditions are shown in Fig. 8. These observations show the strikingly different antecedent soil moisture conditions in these two years, with dry antecedent conditions in 2013 and wet antecedent conditions in 2014. It is well understood (Gray and Landine, 1988, Ireson et al., 2013, Coles et al., 2016) that the infiltration capacity of frozen soils depends strongly on the antecedent soil moisture. When wetter soils freeze, ice-filled pores develop, giving the soil a relatively low infiltration capacity. Drier soils (or more specifically, soils where the largest significant pores, which may be macropores, remain air filled) can maintain a high infiltration capacity when frozen and the snowmelt infiltration can be significant, whilst runoff may be negligible, as in 2013.

**3.3 Growing period**

Figure 9 shows the observed water budget for the growing period in 2013 and 2014. There is a large, systematic bias in the raw net precipitation, which is caused by the energy balance closure correction for the eddy flux measurement of evaporation. This clearly highlights the importance of making this correction, whereas the net precipitation was actually positive prior to adjusting for the lack of energy balance closure. In both years, we see $E$ exceeding $P$ and the soil moisture being drawn down over the summer, highlighting the importance of snowmelt for sustaining agriculture in this region. In both years, the cumulative bias-adjusted net precipitation is within the confidence intervals of the change in $\Delta S_V$. It should be noted that the error bars in Figure 9 indicate the 95% confidence intervals of the mean change in soil moisture between two adjacent measurement dates, whereas the larger uncertainty value shown for $\Delta S_V$ in Table 1 is obtained from the confidence intervals around the mean seasonal moisture change. In 2013, soil moisture was supplied through snow melt (see above) and was progressively depleted by evapotranspiration through the summer months, with minimal rainfall inputs until a large event in late September (Fig. 2). At the end of the season, net precipitation had exceeded the mean change in soil moisture storage, leaving a residual of -40±25 mm. This suggests there was some additional loss of water from the domain, likely as drainage, but that the magnitude of the drainage flux cannot be reliably quantified from these measurements (i.e. it is within the error bars). In 2014, due to significant rainfall in May and June, soil moisture continued to increase until July. In June, net precipitation exceeds the change in soil moisture storage, meaning it is likely some runoff or drainage occurred in response to the large rainfall events. This may explain the observed water table response in the summer of 2014, shown in Fig. 2. In the fall there tends to be less uncertainty in the soil moisture change (due to less spatial variability), and the cumulative net precipitation ends up lower than the change in soil moisture storage, with a residual of 32±18 mm.

While we cannot directly measure soil drainage, we have measured the water table response, 3-5 m below ground, which gives some qualitative indication of the timing and relative amount of soil drainage. In 2013, the water table in piezometer 1 (P1) rose steadily during the growing period, though the amount of rise was small, ~ 20 cm (Fig. 2e). This is consistent with our water balance based estimates of soil drainage, which suggested there may have been a small excess in net precipitation through the summer. Combined with the absence of observation of runoff or stream discharge, we can be reasonably confident that this excess did go to soil drainage. In 2014 the water table in P1 rose higher, ~ 50 cm rise (Fig. 2e), implying that there was significantly more soil drainage this year, which is again consistent with our water balance in June of 2014. In 2014 two additional piezometers (P2 and P3) were available, including one piezometer (P3) located below a topographic depression. The response of these three piezometers is shown in Fig. 10, along with the ground surface elevation. The water table below the depression rose markedly more than in the upland piezometers, and peaked much earlier (July). This is consistent with the depression focused recharge mechanism that has been proposed for these environments (Hayashi et al., 2003), though it is important to note that the dominant recharge signal in this study appears to originate from rainfall and not snowmelt.

## 4. SUMMARY AND CONCLUSIONS

In this study we have used a suite of relatively standard instrumentation to explore the field scale water balance. Our findings are of practical importance for those wishing to measure the field scale water balance, to interpret water balance residuals or use such field scale observations to calibrate/validate models. Due to the local climate, the year is split into three periods, each summarized in the following paragraphs.

During the winter, i.e. the snow accumulation period, we were unable to close the water balance because we did not directly measure the fluxes of blowing snow or upward soil moisture redistribution, both of which are shown to be significant. The results of this study emphasize three practical points that should be considered before using similar data to constrain or validate hydrological models: (1) it is critical that solid precipitation records should be adjusted to remove bias due to wind-induced undercatch; (2) a well-timed snow survey to reveal the peak SWE in the snowpack before melt can capture the pre-melt spatial variability, it can help negate the requirement to capture blowing snow and sublimation, and can minimize uncertainties in the measured solid phase precipitation; and (3) measuring soil moisture prior to melt, if possible, can be extremely valuable to partition soil moisture increases due to over-winter upward redistribution, from increases due to infiltrating snowmelt.

The snowmelt period in the Canadian prairies, as illustrated by this field study, strongly dominates the subsequent hydrological processes. A fundamental challenge is to predict how the melting snowpack will be partitioned between runoff and infiltration, which is a strong determinant of flood risk and soil moisture availability. Our observations demonstrate nicely how SWE alone is a poor predictor of runoff, and are consistent with past studies that have highlighted the importance of antecedent soil moisture in generating runoff.

From a water balance perspective, the growing season was the least problematic in this study. Here, the important question for agricultural production is how much water is available for use by plants. A simple vertical water balance (rainfall minus evaporation) seems to adequately explain the changes in moisture. However, a significant admonition here is that the errors in our water balance were substantially decreased by forcing energy balance closure which caused a sizeable increase (35-39%) to the evaporation amount. This approach is not universally accepted, but in

this instance it seems to be warranted. There were likely small amounts of drainage in both years, more in 2014 due to large rainfall events, but even neglecting these does not result in a large error in the water balance, which is to say they are small compared with the observational errors in $P$, $E$ and change in soil moisture storage. Our naïve 1D water balance approach is likely acceptable for tracking short term (e.g. one season) hydrological processes, with applications for agricultural water usage through the growing season. However, longer term groundwater recharge and solute transport processes (e.g. salts and nutrients) are driven by fluxes that may be much smaller than these residuals, but are important over 10s to 100s of years. In this case, we are limited by the accuracy of our precipitation and evapotranspiration estimates, and other methods, such as chemical tracers, might be useful to help quantify these uncertain fluxes.

## ACKNOWLEDGEMENTS

The authors thank Amber Peterson, Bruce Johnson, Dell Bayne, Brenda Toth and Erica Tetlock who participated in field data collection, as well as Craig Smith and Daqing Yang (National Hydrology Research Centre, Environmental Canada) for providing an unpublished empirical relationship for the Geonor gauge correction. Financial support was provided by the Canada Excellence Research Chair in Water Security, University of Saskatchewan.

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

| Seasons | Observations [mm] | | | | Residual [mm] |
|---|---|---|---|---|---|
| | $P$ | $E$ | $\Delta S_s$ | $\Delta S_v$ | $R$ |
| Snow Accumulation '13 (*Nov. 1 - Apr. 8*) | 72±2 | 14±1 | 97±10 | 24±11 | -63±15 |
| Melt '13 (*Apr. 9 - May 6*) | 10±1 | 16±1 | -97± | 54±26 | 37±28 |
| Grow '13 (*May 7 - Nov. 6*) | 220±6 | 255±2 | 0 | -75±24 | 40±25 |
| Snow Accumulation '14 (*Nov. 7 - Apr. 2*) | 72±4 | 10±1 | 62±7 | 13±5 | -13±10 |
| Melt '14 (*Apr. 3 - Apr. 24*) | 33±2 | 15±1 | -62±7 | 30±28 | 50±29 |
| Grow '14 (*Apr. 25 - Oct. 22*) | 281±6 | 343±3 | 0 | -30±17 | -32±18 |

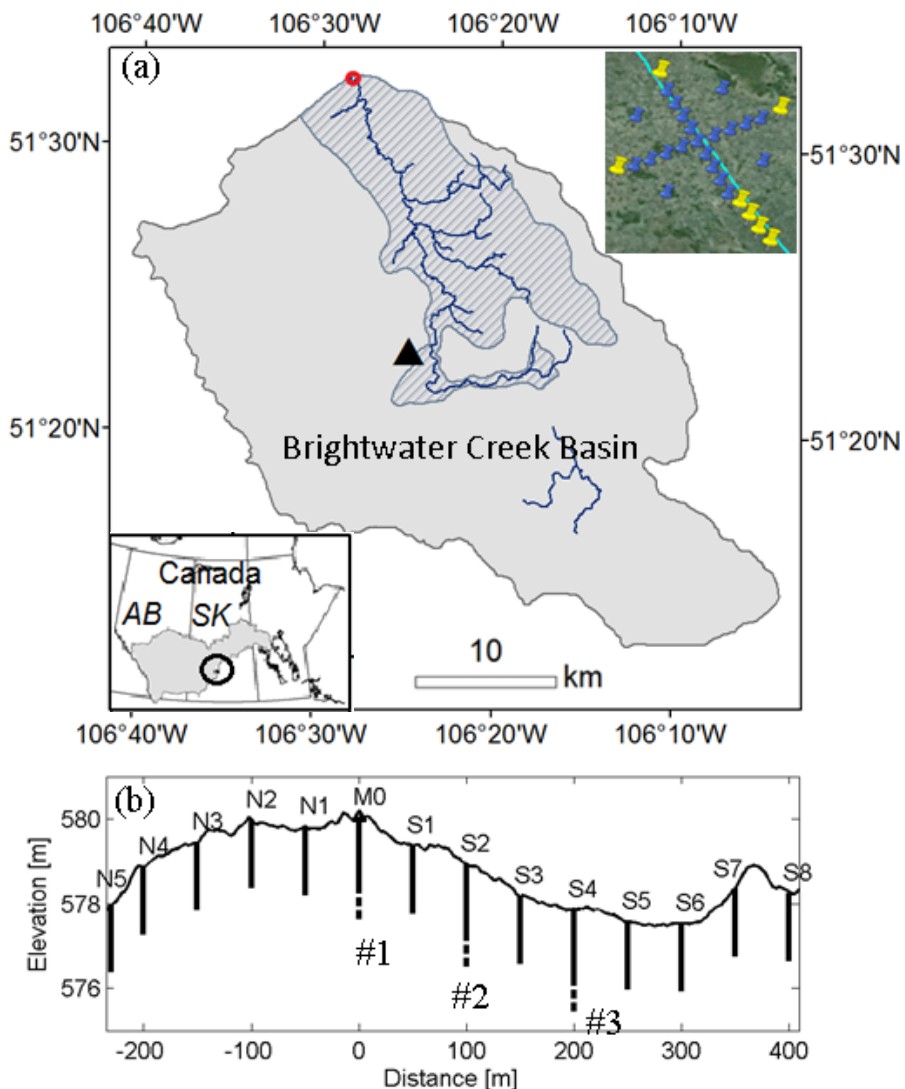

Figure 1 Brightwater Creek sub-basin in Saskatchewan River basin (effective drainage area shown
by hatching) and locations of measurements. (a) Flux tower (triangle) and the schematic
distribution of neutron monitoring locations (right upper corner); Red circle: discharge
measurement location. (b) Instrumentation along the long transect in the (a): neutron probe access
tubes (N5, N4, N3, N2, N1, M0, S1, S2, S3, S4, S5, S6, S7, S8), snow survey (the same location
as the tubes), and three 6-m piezometer boreholes (#1, #2 and #3).

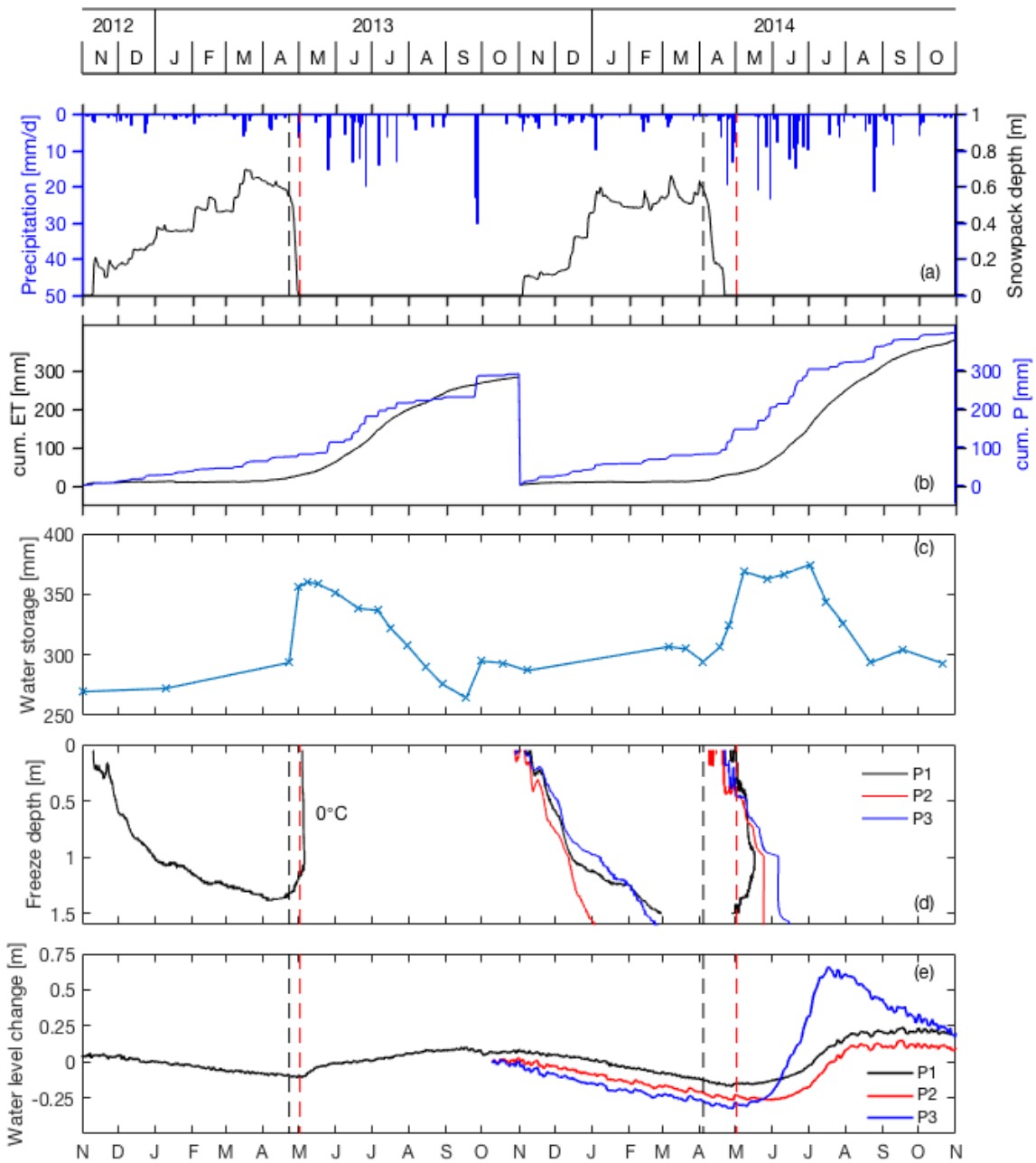

Figure 2 Fluctuation of the major variables in vadose zone hydrology during the years of 2013 and 2014. (a) Precipitation and snowpack depth measured at the flux tower. (b) Yearly cumulative change of evapotranspiration ($E$) and precipitation (P). (c) Average soil water storage change in shallow vadose zone (neutron probe data). (d) Surficial soil freezing above groundwater table at three locations (P1, P2 and P3). (e) Seasonal fluctuation of groundwater level. Black and red dashed lines are the start and end of the snowmelt periods.

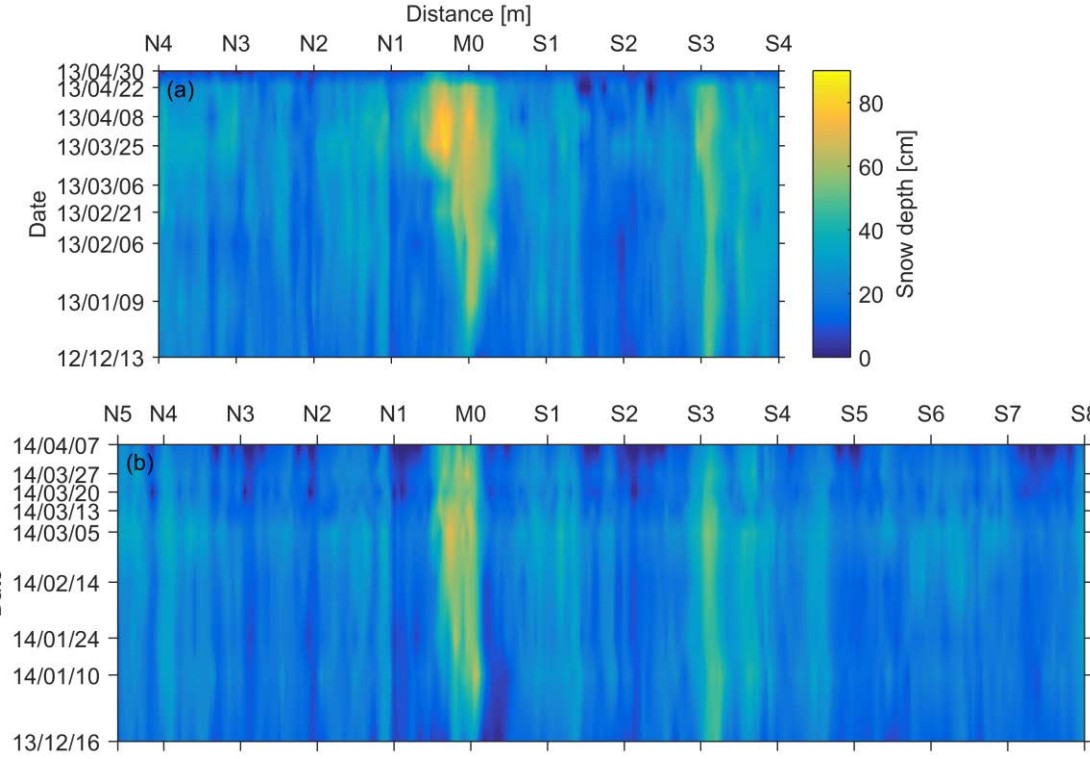

2 Figure 3 Spatio-temporal variation of snowpack depth along the snow survey transect in 2013 (a)

3 to 2014 (b).

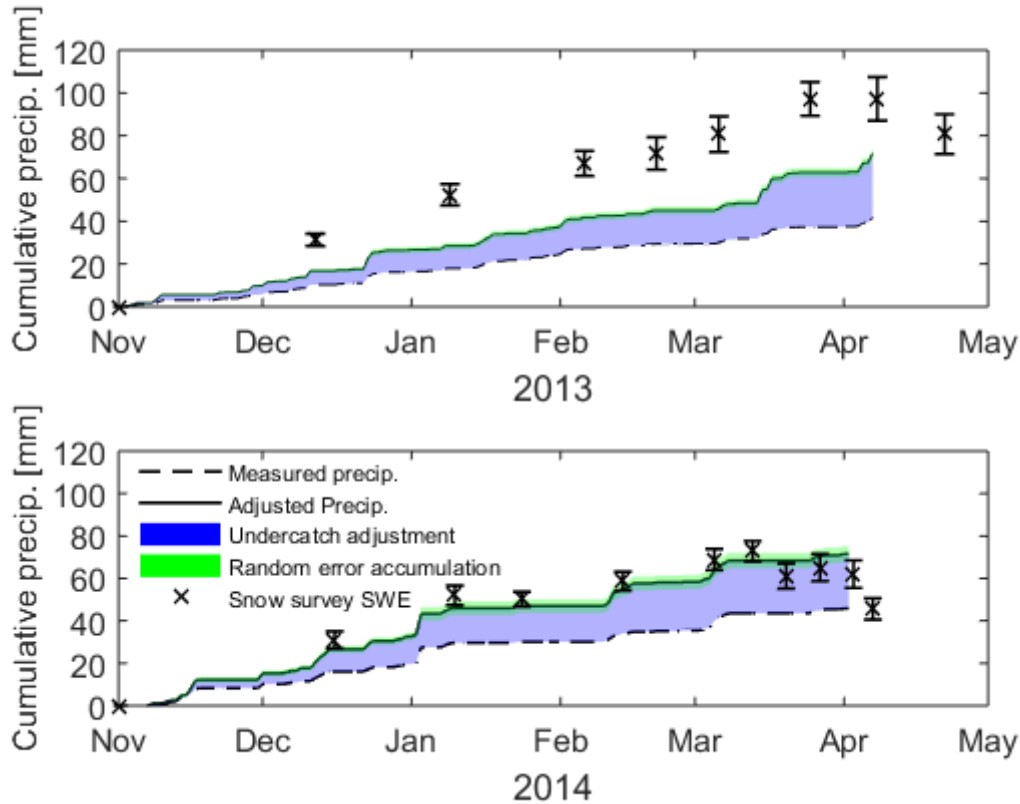

Figure 4 Comparison of measured solid precipitation, bias-adjusted precipitation, and measured snow on the ground during the snow accumulation period in hydrological years of 2013 (a) and 2014 (b). Note the error bars indicate the 95% confidence intervals of the measured SWE.

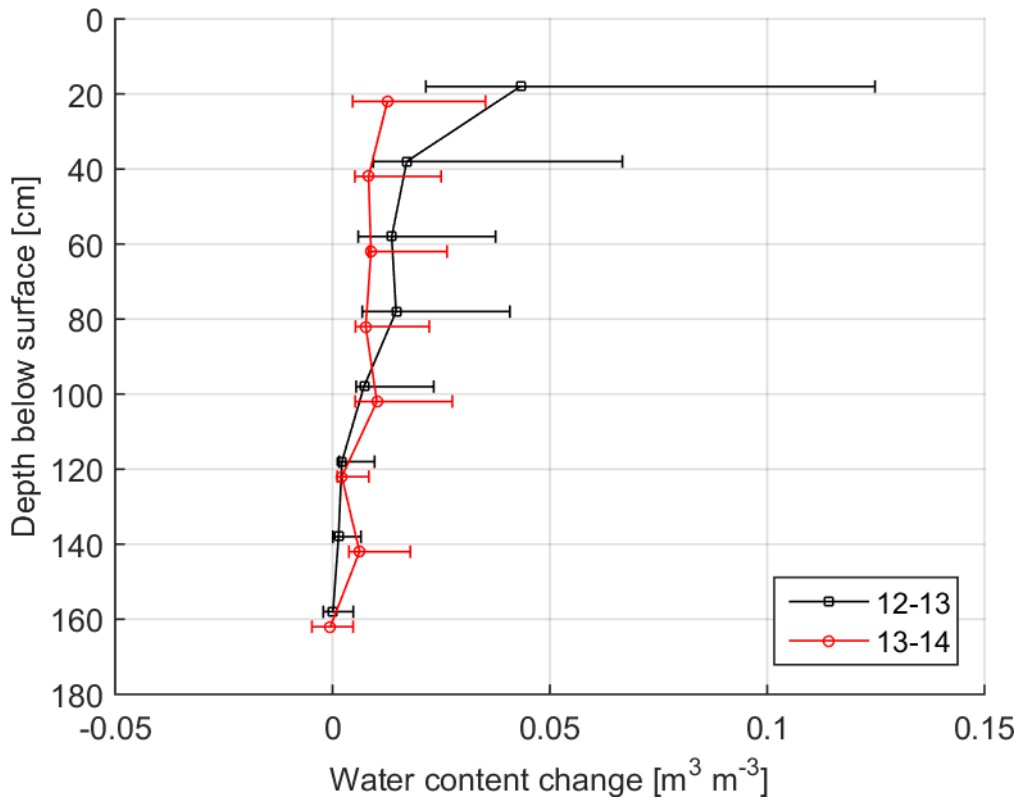

Figure 5 Over-winter change in water content with depth below ground. Symbols indicate the
mean over all measurement locations, and bars indicate 95% confidence intervals. In water year
2013, the pre-freeze measurement was taken on November 1st, 2012, and the pre-snowmelt
measurement on April 22nd 2013. In water year 2014, the pre-freeze measurement was taken on
November 7th, 2013, and the pre-snowmelt measurement on April 3rd, 2014. Note that the
measurements were taken at 20 cm depth intervals, but are plotted here as offset ± 2cm for clarity.

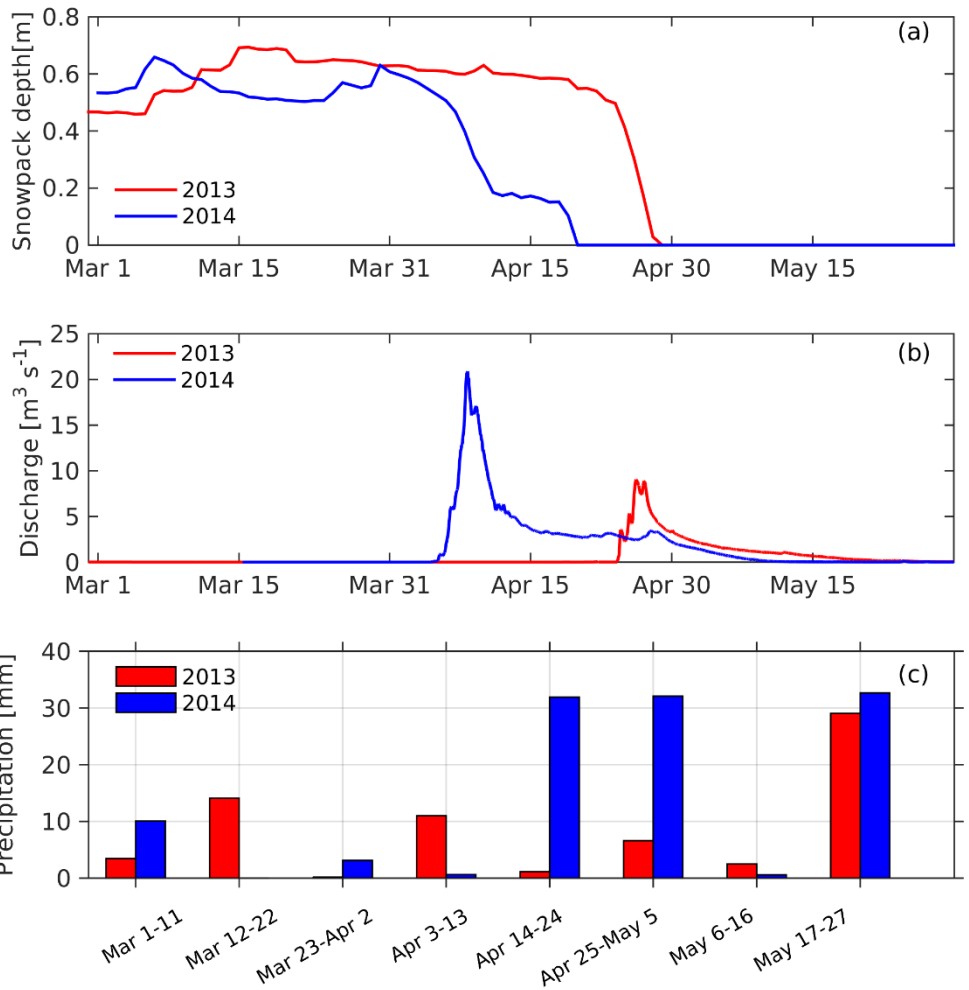

2 Figure 6 Contrasting snowmelt processes in 2013 and 2014. (a) Snowpack depth. (b) Hydrograph

3 of the Brightwater Creek. (c) Snowfall and rainfall with 10-day interval during melt period.

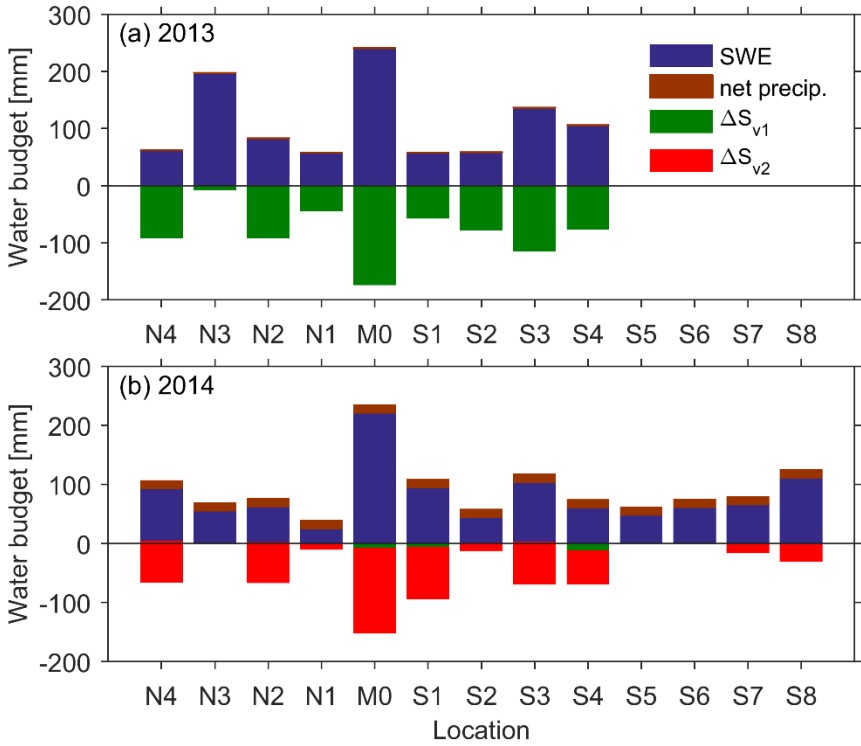

Figure 7 Spatio-temporal variations of soil water storage change in the shallow vadose zone along the Neutron Probe transect during the melt period (between black and red dashed lines in Fig. 2). *SWE*: measured maximum snow storage; *net precipitation:* cumulative difference between precipitation and evapotranspiration; $\Delta S_{v1}$ and $\Delta S_{v2}$: soil water storage change during post-snowmelt and post-thaw periods.

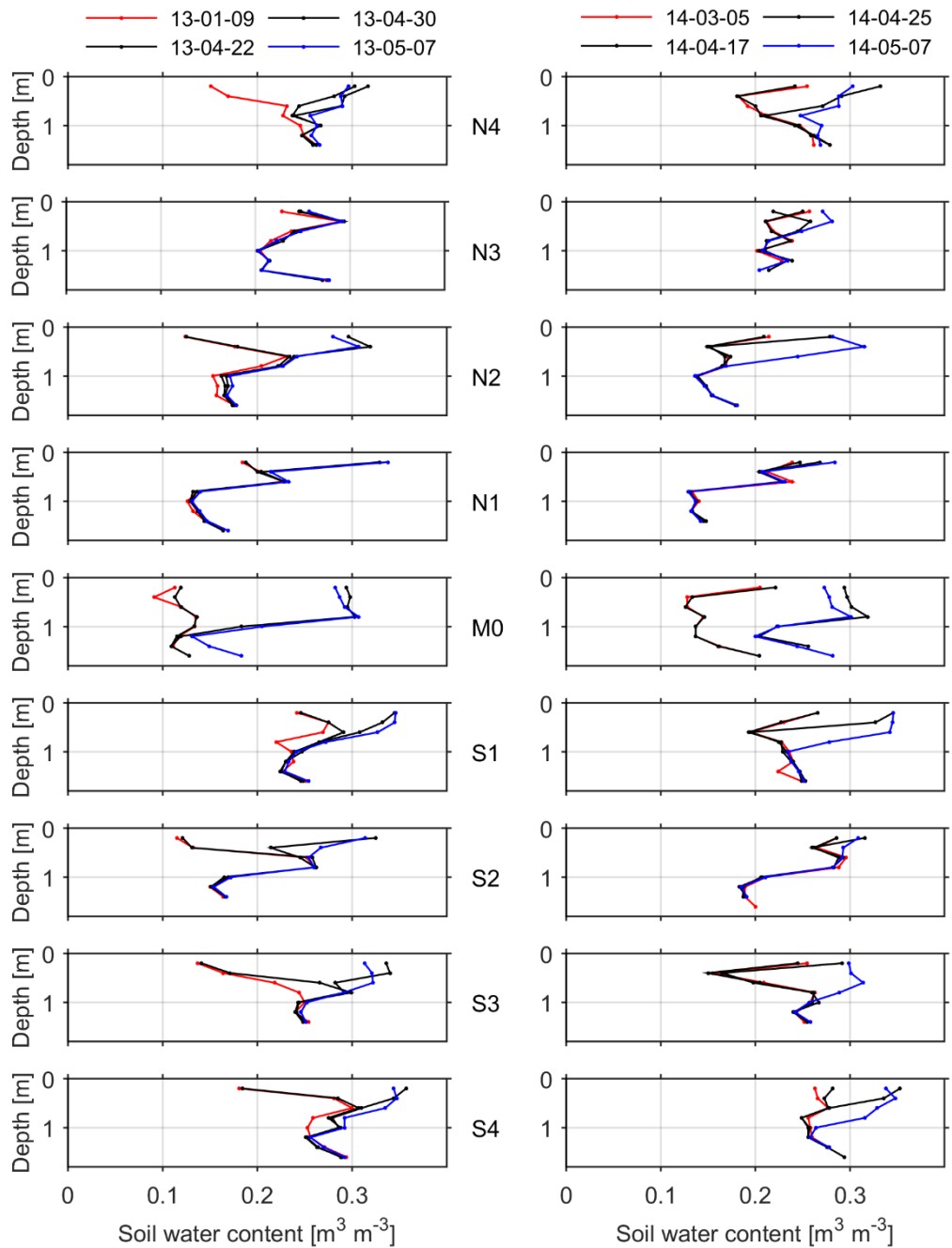

Figure 8 Spatio-temporal variation of water content in shallow vadose zone at different locations

(Fig. 1b) along the Neutron Probe reading transect during the pre-melt (red), post-snowmelt (black)

and post-thaw (blue) in 2013 (left panel) and 2014 (right panel).

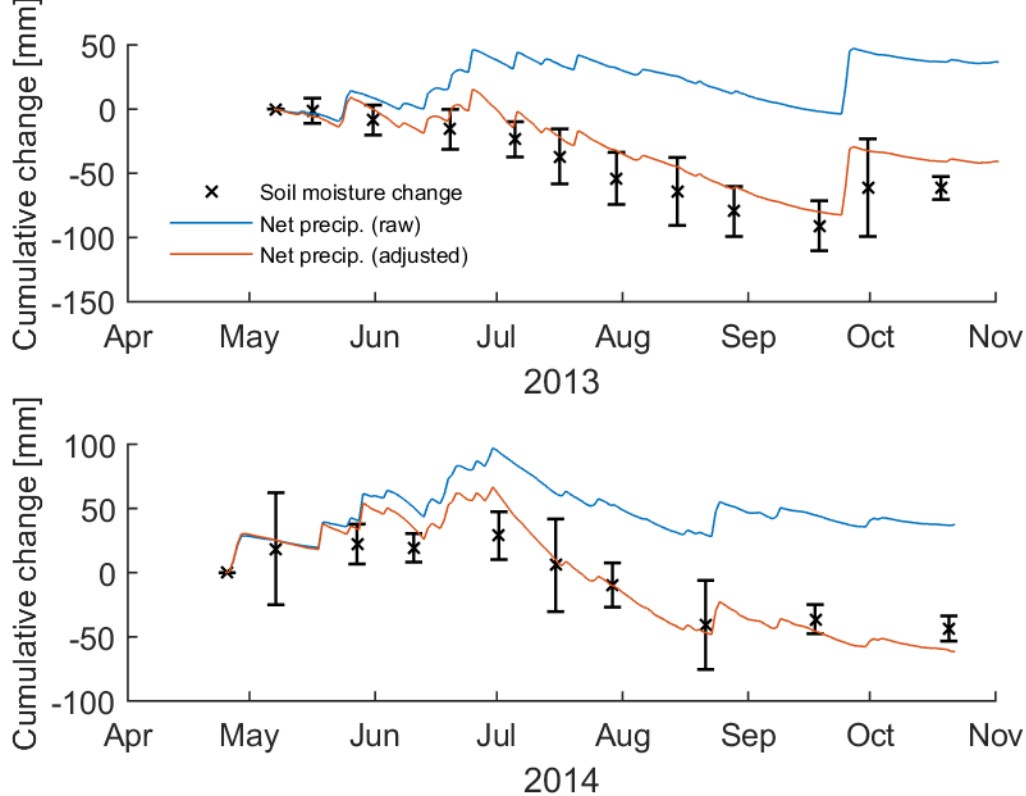

Figure 9 Comparison of net precipitation (P-E), bias adjusted (energy balance corrected) net precipitation, and cumulative changes in soil moisture during the growing period of hydrological years (a) 2013 and (b) 2014. Note that the error bars indicate the 95% confidence intervals of the mean change in soil moisture between adjacent sampling dates.

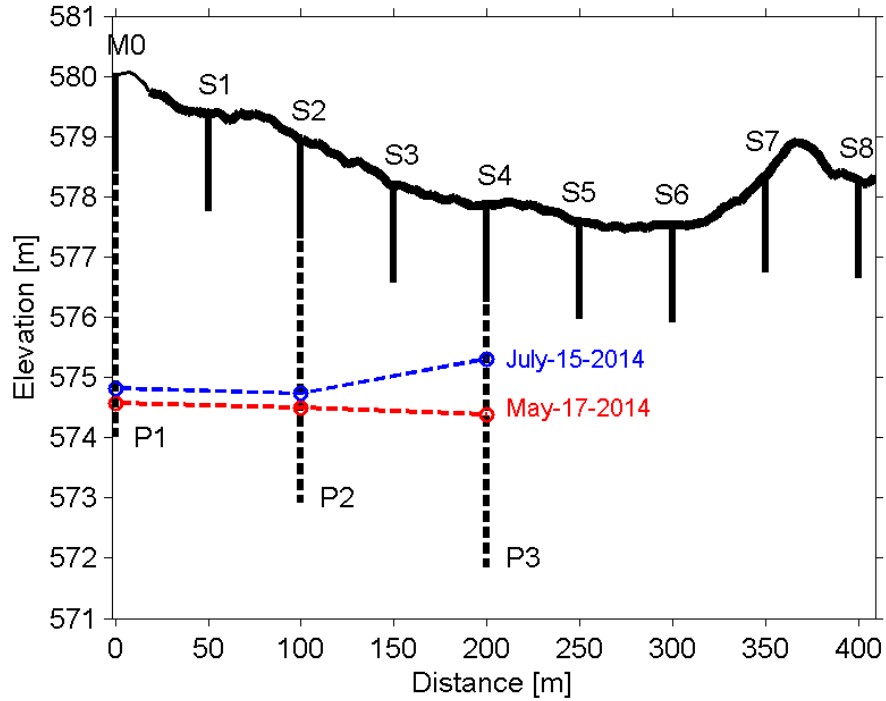

2      Figure 10 Groundwater rise along the slope during the early growing period in 2014.