# Peer review of "Field-scale water balance closure in seasonally frozen conditions"

_Hydrology and Earth System Sciences, 2016_

## Referee Comment (RC1) · Anonymous Referee #1 · 7 Jul 2016

In this manuscript the authors describe a study that confirms care must be taken when using residuals of the water budget to estimate hydrological fluxes. This is a particular problem in cold regions where measurement accuracy is typically lower. Furthermore, the results as presented suggest such residuals would be associated with high levels of uncertainty, because the residuals contain all the error associated with the estimates of the other fluxes. This is perhaps not a novel finding on its own. Could I suggest that the manuscript would be much improved if the authors, particularly in the discussion and the summary, focused on the implications of this uncertainty (as is noted in the abstract), or make recommendations of when certain observations could be more or less important to reducing the uncertainty in the residuals? This type of discussion would elevate the paper.

Furthermore, there are three major issues that should be addressed as the manuscript

**proceeds.**

First, there needs to be better analysis of the available data, rather than making assumptions of hydrological behaviour. For example, the effect of antecedent soil moisture on infiltration and runoff.

Second, the authors' conclusions on the influence of antecedent conditions need better justification and discussion. Explore alternative theories (e.g., the rate of snowmelt), and the subtleties of differences in soil moisture. The differences in antecedent moisture were not substantially different between the two study years.

Third, the summary is poor, and needs to be written to provide more impact for the reader. Perhaps this can be addressed as the authors expand the discussion of uncertainty in water budget residuals.

Minor comments:

**ABSTRACT**

Line 8: It is not impossible to measure water budget terms independently in the field. It is difficult, and maybe uncommon. Perhaps rephrase the sentence to say "..... yet in practice it is uncommon to measure every ....."

Line 17: The snow pack does not infiltrate. Rephrase "... melt from the snow pack mostly ....."

**INTRODUCTION**

Page 4 Line 22: I'm not sure you need this sentence on salts as it is tangential to the water budget problem that is the focus of the paper.

Page 5 Line 4: Exactly because of the issues discussed in this paper, I have always shied away from the term "water balance", and preferred to use the term "water budget". The authors might consider using the latter term when appropriate in this manuscript.

Page 5 Line 11: Drainage fluxes while measured at a point in space, and not point measurements, but integrated over an area. They are not measured at a point scale, and this sentence needs to be corrected.

Page 6 Line 10: The figures in the paper are not presented in order. The first figure presented should be numbered one, not five. Furthermore, maybe the content in these couple of sentences should be presented in the results section.

Equation 1: The field scale vertical water budget should also include melt (M).

Page 6 Line 16: Perhaps G should be ".... net drifting snow over the field domain ....." That might be more accurate.

Page 7 Line 12: I disagree that a +2°C temperature is a killing frost. Maybe -2°C; is this what the authors meant?

Page 8 Line 9: Streamflow is typically a rate (m3/s), but here the total volume is presented. That would be yield. This sentence should read " Mean annual yield in the Brightwater ......"

Page 13 Line 14: Again, check the order in which figures are presented.

Page 13 Line 20: Please provide the data that supports this statement that there were similar surface temperatures. Furthermore, it is wrong that wetter soils freeze faster or deeper. Wetter soils are warmer soils because of the energy required to freeze the water content. An alternative hypothesis that the authors do not consider to explain the different soil temperatures was the growth of snowpack development relative to the air temperatures.

Page 14 Line 10: Another flux is vapour migration from the snowpack to the soil. This is what creates depth hoar in the snowpack. Quinton was one of the first to document this.

Page 15 Line 3: It is effective drainage area not contributing area; stay consistent to

the language you used earlier in the paper.

Page 15 Line 8: Showing increases in soil moisture as negative is not intuitive. Perhaps this should be changed.

Page 15 Line 23: I disagree that the soil moisture conditions were strikingly different. They were both near 0.2. This is not enough to explain the different responses.

Page 16 Line 3: These are saturated at this level? What is the porosity?

Page 17 Line 3: if the error is due to unaccounted for soil drainage, why did it not happen in both years? Please discuss.

Page 18 Line 13: As noted in the major comments, the authors need to expand the discussion to include the implications for hydrologists. Also, a good reference to include in this would be Barr et al. (2012).

Barr et al., 2012. Energy balance closure at the BERMS flux towers in relation to the water balance of the White Gull Creek watershed 1999–2009. Agricultural and Forest Meteorology 153: 3-13.

Page 19 Line 7: This should read "..... because we did not measure the fluxes..... "

Figure 2: Could the authors please provide an explanation for the strange soil temperatures in 2014. Also, related to the discussion the authors have a 'post snowmelt' period, it was hard to judge exactly where on these figures that was, so perhaps another vertical line would help.

Figure 6: If the authors had a Geonor precipitation gauge, why not present daily precipitation rather than 10 day intervals?

Figure 7: There was no change in soil moisture post thaw in 2013? And shouldn't net precipitation be negative – see Table 1.

Figure 8: Why are there 4 lines, but only 3 in the legend?

---

## Referee Comment (RC2) · J. Buttle (Referee) · 13 Jul 2016

This is a well-executed study and a generally well-written paper (although I have made suggested edits directly on the ms). However, my main reservation is that it is not presented as a particularly novel study. Much of the work confirms long-standing knowledge of key hydrological processes in the Canadian prairies (e.g. Gray's work on the significance of pre-freeze-up soil water content on subsequent snowmelt infiltration, Pomeroy's work on snow drifting and sublimation, Hayashi's work on depression-focused recharge). What is needed is a refocusing of the paper to emphasize its novel contributions. Such a refocusing should include a complete error analysis of the various water balance components. As it stands, the paper considers the error in water storage simply in terms of the spatial variation in water storage measured at the various neutron probe access tubes. I feel that this is an

overly-simplistic approach to establishing non-closure of the water balance. The paper would also benefit from a clear definition of what the authors mean by "closure" for the values they present in Table 1, as well as a more complete specification of the main goal of the paper. At present, one of the paper's major goals is to "evaluate whether simplifying assumptions can be justified" (presumably regarding the determination of the site water balance). These assumptions should be spelled out in greater detail, and could be stated as testable hypotheses. In light of these issues, I feel that the paper should not be accepted in its present form. Nevertheless, I think it has promise, and that the authors should be encourage to resubmit a revised version of the paper.

Please also note the supplement to this comment:
http://www.hydrol-earth-syst-sci-discuss.net/hess-2016-260/hess-2016-260-RC2-supplement.pdf
* * *

---

## Author Comment (AC1) · 9 Sep 2016

We would like to thank the anonymous Referee for the comments and suggestions. We have prepared a response to each of the reviewer's comments, and have suggested how we will incorporate these suggestions into a revised manuscript.

With regards to the major comments:

*1. "The First, there needs to be better analysis of the available data, rather than making assumptions of hydrological behaviour. For example, the effect of antecedent soil moisture on infiltration and runoff."*

In this paper we do infer hydrological behavior from our interpretations of our field data set. When the data seem to strongly support a well-established hypothesis about behavior, and no alternative hypothesis is obvious, we do not propose new alternatives. This comment is interpreted to be mainly in the context of the impact of antecedent soil moisture on infiltration and runoff in frozen soils, and blends into the reviewer's next point, which we respond to further below.

*2. "Second, the authors' conclusions on the influence of antecedent conditions need better justification and discussion. Explore alternative theories (e.g., the rate of snowmelt), and the subtleties of differences in soil moisture. The differences in antecedent moisture were not substantially different between the two study years."*

The differences in antecedent soil moisture between the two years were profound and shown clearly (we believe) in Figure 8 (there is a problem with the legend in Figure 8, pointed out by the reviewer, that we will resolve, but this shouldn't affect the point of the figure). In a somewhat simplified interpretation, the 2013 moisture content profiles form the shape of a "Y", whilst the 2014 profiles form a "/". The left branch of the "Y" is the antecedent moisture, which in 2013 is markedly lower. That this should promote infiltration in 2013 and promote runoff in 2014 is not controversial in the literature on frozen soils, going back to Don Gray's work in the Canadian prairies, and many subsequent studies. The differences between streamflow response and surface ponding at the site in the different years are entirely consistent with these observations. The stronger criticism of this (made, in fact, by the second reviewer) is simply that we are not saying anything new (we will address this in our response to reviewer 2). In the revised manuscript, we will more clearly highlight the differences in antecedent moisture, and discuss how this strongly influences the meltwater partitioning and the water balance measurement uncertainty.

*3. Third, the summary is poor, and needs to be written to provide more impact for the reader. Perhaps this can be addressed as the authors expand the discussion of uncertainty in water budget residuals.*

We do agree that we could improve our summary in a revised paper and we will include a more comprehensive error analysis on our observations, by estimating error bounds for observations of precipitation (due to under-catch), evapotranspiration (due to energy balance closure), SWE (due to spatial variability) and soil moisture (due to spatial variability – this is already included).

Minor comments
*1. Line 8: It is not impossible to measure water budget terms independently in the field. It is difficult, and maybe uncommon. Perhaps rephrase the sentence to say ".. ... yet in practice it is uncommon to measure every .. ..."*

Although it may, theoretically, be possible to measure every term in the water balance, there are virtually no examples of this in the literature; particularly at field-scales. Since this is a very important point of context for this study, and for hydrological science in general, we stand by our statement that it is "usually impossible" to measure every term of the water budget. In a revised version we are happy to include more discussion around this point. For instance, we feel it may be worth adding the qualifier "directly measure" in our sentence. For example, there is no way to directly measure the drainage flux – there are only indirect ways of inferring this (e.g. based on Darcy's Law, based on water balances or based on tracers).

*2. Line 17: The snow pack does not infiltrate. Rephrase "... melt from the snow pack mostly .. ..."*
Agreed. This will be changed in revised manuscript.

*3. Page 4 Line 22: I'm not sure you need this sentence on salts as it is tangential to the water budget problem that is the focus of the paper.*
Agreed.

*4. Page 5 Line 4: Exactly because of the issues discussed in this paper, I have always shied away from the term "water balance", and preferred to use the term "water budget". The authors might consider using the latter term when appropriate in this manuscript.*
The rational for replacing "balance" with "budget" is unclear to us. "Budget" surely implies the amount of water available for some purpose. This is not what we are discussing in this paper – we are seeking to close all terms of the water balance, or critique our inability to do so. We think then the term "water balance" is appropriate.

*5. Page 5 Line 11: Drainage fluxes while measured at a point in space, [are] not point measurements, but integrated over an area. They are not measured at a point scale, and this sentence needs to be corrected.*
The reviewer's point is unclear here. We stated that "drainage fluxes can generally only be measured at point scales." We are referring here to using either Darcy's Law (with head gradients from tensiometers (or similar), and a parametric model for hydraulic conductivity), or a 1D soil water balance to estimate drainage – both of which would be considered standard methods and both of which are indeed point scale estimates.

*6. Page 6 Line 10: The figures in the paper are not presented in order. The first figure presented should be numbered one, not five. Furthermore, maybe the content in these couple of sentences should be presented in the results section.*
The reviewer is correct and we will remove references to Figures 5 and 8 from this part of the revised manuscript.

*7. Equation 1: The field scale vertical water budget should also include melt (M).*
Snow melt is not a flux term, but is rather an internal transformation (phase change) of water which is already accounted for within the domain over which we are defining our water balance, and hence should not appear in Equation 1. Meltwater may become runoff, which we account for in Eqn 1, or it may infiltrate and appear as a change in soil moisture (which also appears in Eqn. 1). We will clarify the domain and our approach in a revised version.

*8. Page 6 Line 16: Perhaps G should be ".... net drifting snow over the field domain .. ..."That might be more accurate.*
We agree that the inclusion of "net" is useful, and we will revise the text to read "G is net drifting snow entering / leaving the field domain laterally".

*9. Page 7 Line 12: I disagree that a +2C temperature is a killing frost. Maybe -2 C; is this what the authors meant?*
Agreed. This was a typographical error.

*10. Page 8 Line 9: Streamflow is typically a rate (m3/s), but here the total volume is presented. That would be yield. This sentence should read " Mean annual yield in the Brightwater ......"*
The reviewer is correct, and we will change the units of streamflow to m3/year. We prefer to talk about flow rather than yield.

*11. Page 13 Line 14: Again, check the order in which figures are presented.*
This will be corrected in a revised version.

*12. Page 13 Line 20: Please provide the data that supports this statement that there were similar surface temperatures. Furthermore, it is wrong that wetter soils freeze faster or deeper. Wetter soils are warmer soils because of the energy required to freeze the water content. An alternative hypothesis that the authors do not consider to explain the different soil temperatures was the growth of snowpack development relative to the air temperatures.*
The reviewer is correct about the effects of latent heat, which were absent from our discussion. We checked the temperatures more carefully, and we think that in fact differences in air temperature (lower in 2014) explain the differences in the freezing depths. We will either include an additional figure that shows the differences in the surface temperatures, or include this information as a subplot in an existing figure.

*13. Page 14 Line 10: Another flux is vapour migration from the snowpack to the soil. This is what creates depth hoar in the snowpack. Quinton was one of the first to document this.*
We were unaware of the evidence for vapour migration from the snowpack to the soil, as opposed to vapour transport within the snowpack, and vapour migration *from* the soil to the snowpack, both of which are discussed in the literature as causes of depth hoar. If a reference to Bill Quinton's work on this can be provided, we would be happy to include a reference to this process.

*14. Page 15 Line 3: It is effective drainage area not contributing area; stay consistent to the language you used earlier in the paper.*
Agreed.

*15. Page 15 Line 8: Showing increases in soil moisture as negative is not intuitive. Perhaps this should be changed.*
We understand the reviewers point - this was done for visual effect in Figure 7. In a revised manuscript we will present this as a grouped bar plot, with all the terms being positive.

*16. Page 15 Line 23: I disagree that the soil moisture conditions were strikingly different. They were both near 0.2. This is not enough to explain the different responses.*
The pre-melt soil water content is indeed strikingly different in 2013 and 2014 – there is a 5-10 % difference in most of the profiles shown. Moreover, the shape of the profiles is completely different. We will add gridlines to Figure 8 which may make this clearer.

*17. Page 16 Line 3: These are saturated at this level? What is the porosity?*
We did not mean to imply here that the soils were saturated, just that they were relatively wet. We will modify this sentence to read "The permeability and hence infiltration capacity of a frozen soil tends to decrease under wetter conditions, as happened here in 2014."

*18. Page 17 Line 3: if the error is due to unaccounted for soil drainage, why did it not happen in both years? Please discuss.*
This point is discussed in considerable detail in the paragraph following the one in question.

*19. Page 18 Line 13: As noted in the major comments, the authors need to expand the discussion to include the implications for hydrologists. Also, a good reference to include in this would be Barr et al. (2012). Barr et al., 2012. Energy balance closure at the BERMS flux towers in relation to the water balance of the White Gull Creek watershed 1999–2009. Agricultural and Forest Meteorology 153: 3-13.*
This is an excellent suggestion. We are aware of that paper, which is somewhat uncritical in how it applies a water balance. We will expand our discussion making reference to this paper.

*20. Page 19 Line 7: This should read ".. ... because we did not measure the fluxes.. ... "*
Agreed.

*21. Figure 2: Could the authors please provide an explanation for the strange soil temperatures in 2014. Also, related to the discussion the authors have a 'post snowmelt' period, it was hard to judge exactly where on these figures that was, so perhaps another vertical line would help.*

Figure 2(d) provides freezing fronts (0°C), which are interpolated from soil temperature measurements from a series of sensors (Stevens Hydro-probe) in depth. In 2014, there were three profiles with different snowpack thicknesses, which led to different soil freeze-thaw rates. P2 and P3 had thinner snowpack and thicker freezing depth than P1.

Regarding the "post snowmelt" period, we will try to more clearly distinguish this in a revised version of the paper.

*22. Figure 6: If the authors had a Geonor precipitation gauge, why not present daily precipitation rather than 10 day intervals?*

In this figure we present the 10-day accumulation so that is it easier to contrast the difference in precipitation type between the 2 years. The daily precipitation is already shown in Figure 2(a).

*23. Figure 7: There was no change in soil moisture post thaw in 2013? And shouldn't net precipitation be negative – see Table 1.*

During 2013, there was a negligible change in measured soil moisture between our last measurement date during the snowmelt event and a subsequent measurement date when soils had thawed. The purpose of presenting these data in this manner was to contrast these 2 years. In 2014, due to restricted infiltration, any measurable changes in soil moisture were delayed until the soils had sufficiently thawed. In reality, our neutron probe measurements only provide a rather coarse temporal resolution, and melt and thaw cannot be considered as discrete events. However, we feel that our generalizations are useful to contrast the timing of the infiltration processed during these 2 years.

With respect to the net precipitation, indeed, the 2013 number should be negative. This will be corrected in a revised version.

*24. Figure 8: Why are there 4 lines, but only 3 in the legend?*

Good point. This figure will be revised to reflect the actual dates of measurement, while keeping the color scheme to distinguish which hydrologic regime (e.g. pre-melt, post-melt, etc.) each measurement date corresponds to.

---

## Author Comment (AC2) · 9 Sep 2016

We would like to thank the reviewer, Jim Buttle, for taking the time to review our manuscript and for providing constructive feedback.

A number of markups on the PDF of the manuscript were provided by the reviewer, most of which are straightforward corrections which will be adopted in a revised manuscript without further comment here. Non-trivial corrections are addressed below under "Reply to minor comments".

By our reading, the reviewer has three major comments that we need to address:

*1. Refocus the paper to emphasize the novel contributions*
The reviewer correctly notes that the processes that we describe in our paper are not novel process descriptions/explanations. The contributions of this paper are: i) that we explicitly look at these processes and the water balance on an integrated scale, namely the field scale, root zone, by considering field-integrated observations; ii) that we seek to test existing hypotheses against our dataset at this scale (and essentially find that we corroborate these hypotheses); and iii) we seek to clearly show the limitations of using water balance residuals terms as either field based estimates of fluxes, or as a means of validating models (a point which is perhaps not novel in general, but is routinely violated in practice, including in the academic literature). On the latter point, we show that it is useful, in the Canadian prairies, to consider three separate time periods, (snow accumulation, melt and summer), with some periods raising less difficulties for water balance residuals than others. The single greatest problem in terms of quantifying the water balance, is partitioning of melt between runoff and infiltration, and we feel it is valuable to point this out. We do agree that these contributions are not as clearly described as they could be, and we will endeavor to improve our Abstract and Summary and Conclusions sections.

*2. Present a complete error analysis of the various water balance components, and define what is meant by closure*
In terms of closure, we do have a definition on page 3, line 4: "In the current paper, we define the problem of water balance closure as that of independently quantifying each term in the water balance equation, such that the changes in storage within a specified domain and over some time interval are adequately balanced by the net fluxes into/out of that domain over the same time interval." In our analysis we try to be clear about terms that are measured, and terms that are not, and we refer to the latter as closure estimates. We will try to include a more comprehensive error analysis on our observations, by estimating error bounds for observations of precipitation (due to under-catch), evapotranspiration (due to energy balance closure), SWE (due to spatial variability) and soil moisture (due to spatial variability – this is already included).

*3. Present our simplifying assumptions more clearly, perhaps as testable hypotheses.*
We think our point wasn't really to make simplifying assumptions, but to rather explain where observations are unable to capture a particular process, and highlight this as a problem – the biggest of these being the partitioning of melt into runoff and infiltration. The problem then is not coming up with testable hypotheses, it is coming up with ways to test existing hypotheses. We hope this will be clearer in a revised manuscript.

***Reply to minor comments***

p. 11, line 19: The length of the piezometer screens was 33 cm. The piezometers were 5.5 m long and the water table was no deeper than around 4 m below ground. In this situation, we expect the water table elevation and the potentiometric surface at the mid-point of the screened section to nearly coincide. We will further clarify in a revised manuscript.

p. 13, line 21: We agree that the surface temperature data is important (also Review 1 raised this) and we will include this data in a new figure, or as a subplot in an existing figure.

Section 3.3 contained various awkward phrasing that we will improve.

Table 1 labelling will be improved for clarity, and the associated discussion in the text will clarify the rational for the columns labelled "closure estimates [mm]".

Figure 3 and Figure 9 have a caption title that starts "Non-closure of the vertical field-scale water balance…" and perhaps this phrasing leads to confusion. We will delete "Non-closure of" from the start of these captions. The broader point about what we mean by this was discussed above.

Figure 8 – a concern regarding the legend that was also raised by Reviewer 1 and will be resolved in revision of the paper.

---

## Author Response (AR2)

**Reply to reviewers and Editor**

July 11, 2017

"Field-scale water balance closure in seasonally frozen conditions" by Pan et al.

We are grateful for the further reviews and comments on our paper, which have flagged up some minor errors in the text, and allowed us to (hopefully) better clarify the overall purpose of this study.

**Reviewer #1**

The first point of the reviewer is regarding measured and unmeasured terms in the water balance. The reviewer states "Perhaps a better experimental design would have been to attempt to measure water budget terms in Eqns. 1-3 and 5 and then compare those with the 1D water budget equation." We agree that this is a better way to frame the overall objective - this is essentially what we intended originally. Note though that we have only measured those terms that can be measured in a relatively simple, routine manner. For example, there is no simple way to directly measure blowing snow that is routinely applied. There is likewise no simple way to directly measure lateral subsurface flows. We have revised the final paragraph of the introduction to address this point.

The second point of reviewer #1 is regarding the sign of the water balance residuals. We had defined the water balance residual, R, correctly in Eqn 5, such that a positive residual might be explained by a positive amount of runoff (i.e. the reviewer's point). Unfortunately, we had an error in the sign in Table 1, and an error in the symbol (T not R). In the revised corrected table it should be clear there is a large positive residual associated with runoff in the melt seasons. We are grateful for the reviewer spotting this error.

The third point is that the increased runoff response in 2014 seems "disproportionate" to the increase in antecedent storage. The reviewer suggests that looking at the storage deficits might help. We contend, however, that this is not the right approach. It is not storage capacity that limits infiltration in frozen soils - it is the infiltration capacity that results from the reduction in K in the frozen pores. A small addition of water can freeze and block the pore space, which indeed is expected to have a disproportionate, or highly non-linear, impact on runoff. This phenomenon is pretty well understood, and we have added more supporting references to the discussion of this in section 3.2.

**Reviewer #4**

Reviewer 4's main concern (expressed in paragraphs 2, 4, 7 and 8 of his/her 8 paragraph comments) was that the objectives of the paper were unclear - why are we doing this - and also the findings were "overstated". We conceded to both points, and hope that the revised final paragraph in the introduction and the revision to the abstract, conclusions and changes elsewhere, have addressed this concern. Living in the Canadian prairies we do feel that our hydrology is markedly different from most other places, not without good reasons, but in our zeal to explain this we may have been hyperbolic. Our conclusions are largely practical, and we now state this very explicitly in the first paragraph of the conclusions. We did not intend to imply that we are the first to state that precipitation undercatch is important, for example. The point of the paper is to provide a number of important considerations for those measuring water balances, interpreting water balance residuals (people do this a lot, implicitly) and using observations to constrain models. These points are not individually novel, but we do think they offer valuable guidance in an area where, in our experience at least, best practice is frequently not followed.

Examples of how we have "tempered" our "hyperbole" include: replace "exceedingly complex" with "complex" (p. 4); and deleting the offending sentences "As a result of the aforementioned complexities, prairie water balance is subject to spatial variability of precipitation inputs, water storage characteristics, and evaporation patterns. Because these hydrological characteristics do not necessarily vary at the same scales, the challenge of properly quantifying the water balance becomes daunting."; removed "albeit unusually comprehensive" in the conclusions.

We have removed the bullet points on page 7.

Reviewer 4 commented that the error assessment was overstated. We used the error analysis to try to establish whether water balance residuals might be down to instrumentation or sampling errors, or a missing process. This is hopefully stated more clearly in the introduction now. We used the simplest approach we knew of, with simplifying assumptions about where we think the dominant error comes from for a particular measurement, and including only one source of error for each measurement (i.e. if sampling errors are expected to be larger than measurement precision, we ignore measurement precision errors, and vice versa). We did not intend to make any claims that we are doing anything novel in the area of error analysis, and we're not sure where the reviewer finds us overstating this.

The reviewer raises interesting questions about processes during the winter, which resulted in deeper freezing in 2014. The question is, why did the soil freeze deeper when it was wetter? We agree that this is a very interesting problem and agree that latent heat is an important consideration that we did not mention. However, this is a complex issue on the periphery of the objectives of this paper, so we do not wish to get into a detailed discussion of this here. We have deleted the comment about thermal diffusivity and now state "
[revised manuscript text omitted]

---

## Author Response (AR3)

**Reply to Editor**

July 24, 2017

"Field-scale water balance closure in seasonally frozen conditions" by Pan et al.

Thank you for pointing out these last few minor corrections. We have addressed each of them as described in the following text:

1. This text on P5 starting L21 seems misplaced "If the non-measured terms are negligible, i.e. within sampling and instrumentation errors, we are left with a 1D water balance where net precipitation (precipitation minus evapotranspiration) is adequately well balanced by changes in storage only. If the non-measured terms are non-negligible, there will be a non-zero water balance residual. We interpret these residuals for the different seasons."
It seems to me that it is not possible for the reader to understand this without seeing the water balance equation and knowing what was measured and unmeasured. Please revise the last 5-6 lines of the Introduction so that it is self-contained.

**Response: In order to avoid confusion for the reader, we have simply removed this text. It was misplaced, and was not critical to this paragraph.**

2. Coles et al is a discussion paper that was not accepted for publication. Therefore it is not a reliable source and should be removed.

**Response: This reference has been removed.**

3. Table 1, Melt '13: The 97+/- value is incomplete. Should be 97+/- 10 ?

**Response: Yes, the uncertainty value was omitted. This has been corrected.**

[revised manuscript text omitted]